# Effectiveness of an Early Skin-to-Skin Contact Program for Pregnant Women with Cesarean Section: A Quasi-Experimental Trial

**DOI:** 10.3390/ijerph20105772

**Published:** 2023-05-09

**Authors:** Yumiko Igarashi, Shigeko Horiuchi, Beatrice Mwilike

**Affiliations:** 1Graduate School of Nursing, St. Luke’s International University, 10-1 Akashi-cho, Tokyo 104-0044, Japan; igarashi.yumiko.t3@slcn.ac.jp; 2School of Nursing, Muhimbili University of Health and Allied Sciences, Dar es Salaam P.O. Box 65004, Tanzania; beatricemwilike@yahoo.com

**Keywords:** breastfeeding, cesarean section, infant health, kangaroo-mother care method, Tanzania

## Abstract

Objective: This study aimed to clarify the effectiveness of early skin-to-skin contact (SSC) after a cesarean section (CS) program. Methods: An “early SSC after CS” program was implemented at a tertiary care hospital in Tanzania. A non-equivalent group design was used. A questionnaire was used to collect data on exclusive breastfeeding, breastfeeding intention, Birth Satisfaction Scale—Revised Indicator (BSS-RI) score, perioperative pain with a visual analogue scale, and infant hospitalization for infectious diseases and diarrhea at 2–3 days postpartum. Follow-up surveys were conducted until 4 months postpartum regarding exclusive breastfeeding, breastfeeding intention, and hospitalization of the infants. Results: This study involved 172 parturient women who underwent CS, with 86 in the intervention group and 86 in the control group. The exclusive breastfeeding rates at 4 months postpartum were 57 (76.0%) in the intervention group and 58 (76.3%) in the control group, with no significant difference. The BSS-RI score was higher in the intervention group (7.91, range 4–12, SD 2.42) than in the control group (7.18, range 3–12, SD 2.02) (*p* = 0.007) for women who underwent emergency CS. The survival probability for infants hospitalized owing to infectious diseases, and diarrhea was significantly higher in the intervention group (98.5%) than in the control group (88.3%) (χ^2^ = 5.231, *p* = 0.022) for multiparas. Conclusion: The early SSC after CS program showed a positive effect on the birth satisfaction of women undergoing emergency CS. It also reduced the incidence of infants hospitalized owing to infectious diseases and diarrhea for multiparas.

## 1. Introduction

Along with developments in medical technology in recent years, the number of cesarean section (CS) births has also increased globally. Notably, the global CS rate of 12.1% in 2000 increased to 21.1% in 2015 [1]. The World Health Organization (WHO) projects this percentage to further increase over the coming decade, with nearly a third (29%) of all births likely to be via CS by 2030 [2]. WHO has also shown that increases in CS rates of up to 10–15% at the population level are associated with decreases in maternal, neonatal, and infant mortalities. Above this level, increasing the CS rate is no longer associated with reduced mortality [3]. However, CS rates remain high in some countries. We speculate that societal changes usually underlie the increasing CS birth rates. This is in line with a previous study wherein changes in maternal characteristics and professional practice styles, increasing malpractice pressure, as well as economic, organizational, social and cultural factors, were all implicated in this increasing CS birth trend [4]. The countrywide CS rate in Tanzania is not so high at 5.9%. However, Dar es Salaam, which is the largest city and financial center of Tanzania, has the highest CS rate at 17% [5].

Furthermore, various effects of CS on mothers and infants have been reported, particularly regarding breastfeeding. Previous studies have shown that CS delays breastfeeding initiation and increases mixed nutrition [6,7]. Breastfeeding has many advantages, such as acquiring immune function in the newborn. It also prevents and protects against diarrhea in infants under 6 months, particularly infants in exclusive breastfeeding [8]. The risk of infection-related mortality was reported to be 8.66 times higher in 0- to 5-month-old infants without breastfeeding (RR 8.66) than in 0- to 5-month-old infants with exclusive breastfeeding [9]. Breastfeeding should be promoted in developing countries where infection is the cause of common infant mortality and clean water is difficult to obtain.

Skin-to-skin contact (SSC) is recommended in the Baby-Friendly Hospital Initiative launched by the WHO and United Nations Children’s Fund (UNICEF) in 1991 as one of the practices that promotes and supports breastfeeding [10]. Early SSC refers to placing a naked newborn baby wearing only a cap and a nap on the mother’s chest and wrapping the newborn’s back and sides with a blanket or towel. SSC does not require any special tools or incur any costs, making it relatively easy to implement. WHO recommends starting SSC immediately after birth and continuing it uninterruptedly for at least 1 h for all women or until the first breastfeeding [11]. According to a systematic review and analysis of the effects of SSC on exclusive breastfeeding, the duration of exclusive breastfeeding from discharge to 3 months was OR = 2.47 (95% CI: 1.76–3.48; *p* < 0.001), and that from 3 to 6 months was OR = 1.71 (95% CI: 1.05–2.78; *p* = 0.030) compared to without early SSC [12]. Additionally, early SSC is recommended for all babies, regardless of newborn weight, gestation weeks, birth method, or clinical status, including cesarean-born babies [10].

In Tanzania, the mother and infant can usually be together immediately after vaginal delivery; however, in the case of CS, they are separated immediately. The next visit of the mother to her baby is 1 or 2 days postpartum, thus delaying the first breastfeeding. Compared with vaginal delivery, CS was associated with an adjusted prevalence ratio for the early initiation of breastfeeding at 0.24 (95% CI 0.17–0.33) in Tanzania after adjusting for the following confounding factors: pregnancy planning, birth weight, region of residence, sex of child, mother’s age at birth (in years), mother’s education, birth order, number of antenatal visits, maternal tobacco use, place of delivery, household wealth quintile, mother’s occupation, distance to health facility, and urban/rural residence [13]. WHO recommends that the first breastfeeding be performed within 1 h after birth and that exclusive breastfeeding be undertaken for 6 months after birth. However, the proportion of exclusive breastfeeding at 4 to 5 months postpartum in Tanzania is only 26.6% for both vaginal delivery and CS combined [5].

It has also been shown that CS newborns had less SSC immediately after delivery and less breastfeeding within 1 h of birth [14]. A previous study has reported that CS-delivered babies had a 98% decrease in the odds (aOR = 0.02, 95% CI: 0.01–0.05) of receiving SSC compared with vaginal-delivered babies [15]. According to the Centers for Disease Control, 70% of US birth facilities implemented SSC for at least 30 min for most mothers and babies within 2 h after an uncomplicated CS. Additionally, 83% of birth facilities reported in 2015 that they practice routine SSC for most mothers and babies for at least 30 min within 1 h of an uncomplicated vaginal delivery. Compared with vaginal delivery, the rate of early SSC during CS is low [16]. In Africa, few studies have verified the effects of early SSC. The effects of early SSC after CS have not been confirmed. Implementation of early SSC after CS is necessary to accumulate more results.

This study aimed to clarify the effectiveness of an early SSC after a CS program on the exclusive breastfeeding rate of parturient women at 4 months postpartum as the primary outcome, as well as exclusive breastfeeding duration, breastfeeding intention, knowledge of breastfeeding and early SSC, perioperative pain, birth satisfaction, number of infant hospitalizations, and probability of infant survival prior to hospitalization as the secondary outcomes.

## 2. Methods

### 2.1. Study Design

We used a non-equivalent group design with post-test, which is a quasi-experimental study. We implemented an “early SSC after CS program” that included not only the care intervention of early SSC but also the advance education of health-care providers and the provision of information to pregnant women and their families. The feasibility of “early SSC after CS protocol,” the predecessor of this program, was verified in 2018 [17].

### 2.2. Outcomes

The primary outcome was the rate of exclusive breastfeeding at 4 months after delivery. According to the Tanzania Demographic and Health Survey, the percentage of exclusive breastfeeding dropped sharply from 4 months postpartum [5]. It was important as a first aim to improve the breastfeeding rate at 4 months. Exclusive breastfeeding as a primary outcome was defined as only breastfeeding within 1 month at the time of data collection, even if milk was used temporarily during the postpartum period. The secondary outcomes were as follows: (1) days of exclusive breastfeeding until 4 months after delivery, (2) mother’s intention to breastfeed at least up to 6 months after delivery at 4 points (1 to 3 days, 1 month, 2 months, and 4 months after delivery), (3) knowledge test of breastfeeding and early SSC at 1 to 3 days after delivery, (4) score of perioperative pain visual analogue scale at 1 to 3 days after delivery, (5) Birth Satisfaction Scale—Revised Indicator (BSS-RI) score at 1 to 3 days after delivery, (6) number of infant hospitalizations owing to infectious diseases and diarrhea at 4 time points; (1 to 3 days, 1 month, 2 months, and 4 months after delivery, and (7) survival probability for infants hospitalized due to infectious diseases and diarrhea or death.

### 2.3. Setting and Study Period

This study was conducted at Muhimbili National Hospital located in Dar es Salaam, Tanzania. The data collection period was from 1 June 2021 to 31 December 2021.

### 2.4. Participants and Sample Size

The study participants were pregnant women undergoing CS and their infants. The maternal inclusion criteria were as follows: (1) all types of CS (both planned and emergency), (2) 18 years or older, (3) 34 weeks gestation or later, and (4) can speak Swahili. The maternal exclusion criteria were as follows: (1) pregnant women who have difficulty communicating owing to severe mental illness, (2) severe anemia, (3) severe preeclampsia, (4) severe hypertension owing to pregnancy, and (5) usage of general anesthesia.

The implementation of SSC after CS was decided after assessing the postpartum condition of the mothers and infants. The infant exclusion criteria were as follows: (1) abnormal respiration, (2) whole-body cyanosis, and (3) hypothermia. If the infant died immediately after delivery, the infant was excluded because data on subsequent breastfeeding could not be collected. If the infant had problems such as hypothermia, dyspnea, and cyanosis, the health-care providers immediately provided treatment.

The sample size was calculated using G*Power (3.1.9.6 for Mac OS 10.7 to 13; Heinrich-Heine-Universität Düsseldorf: Germany, 2020) in reference to a previous study [18]. As for the effect size of the intervention, the difference in exclusive breastfeeding rate with and without early SSC was set at 15%. The significance level was set as <0.05 and the power value as 0.8. The total sample size should be 144 with 72 in each group. Considering a 20% dropout rate, the total sample size was set at 172, with 86 in each group.

### 2.5. Program

The early SSC after CS program consisted of a three-step intervention. First, education about the importance of breastfeeding and early SSC was provided to pregnant women at the antenatal clinic or anteroom of the operating theater using an educational pamphlet. Second, simulation training for health-care providers was conducted. Third, early SCC after CS was implemented.

#### 2.5.1. Education about the Importance of Breastfeeding and Early SSC

As this study included emergency CS, it was difficult to select and educate only pregnant women who could be predicted to have a CS. Thus, education was provided to all pregnant women. An antenatal clinic midwife provided education on breastfeeding and early SSC to all pregnant women who came for a prenatal checkup, using pamphlets. The session took about 10 to 15 min. They were able to take the pamphlet home so that they could discuss it with their families when they returned home.

#### 2.5.2. Simulation Training for Health-Care Providers

First, an early SSC promotion team consisting of three midwives working in the operating room was formed. Before implementing the early SSC after CS program, the operating room midwives, anesthesiology staff, and obstetricians received education for approximately 30 min on the definition of SSC, its significance, effectiveness, rationale, and procedure; and observation points regarding maternal state of consciousness, infant hypothermia, infant respiratory abnormalities, cyanosis, and neonatal resuscitation (Appendix A, Table A1). Education was provided by the investigators or members of the early SSC promotion team. Education for health-care providers was conducted as often as possible to ensure that many staff members could participate. After the implementation of the early SSC after CS program, team members conducted simulation training for health-care professionals twice a month, focusing on practical skills.

#### 2.5.3. Implementation of the Early SCC after CS Program

Early SSC was defined as SSC performed within the first hour after delivery [19]. First, a midwife would check for the following initiation criteria: full consciousness of the mother, no abnormal breathing in the infant, and absence of hypothermia in the infant. Early SSC was scheduled from the time the parturient woman and infant were able to undertake early SSC after CS until they left the operating room. If a midwife determined that SSC was feasible after meeting the initiation criteria, the bare infant was placed on the mother’s chest skin to skin in a vertical or oblique position to the mother’s body. The infant’s face was turned to the side to avoid breathing disturbance, and the infant’s head was wrapped in a warm cloth. Unless medically contraindicated, the parturient woman supported the infant by herself to allow her to touch her baby. During the early SSC, the midwife monitored the conditions of both the mother and the infant according to a checklist of continuation criteria, which were similar to the initiation criteria. The midwife stopped the early SSC if the continuation criteria were not met or if the mother requested discontinuation. If fetal distress (Apgar score ≤ 7 points), abnormal breathing, generalized cyanosis, or hypothermia was observed during the early SSC, it was discontinued.

### 2.6. Delivery Care for Control Group

This study was divided into two phases. In the first phase, education on breastfeeding and early SSC was provided to all pregnant women at the antenatal clinic using an educational pamphlet. A control group was recruited and received current care during CS without early SSC. After the baby was born, the midwife showed the parturient woman her infant, and then took the infant to an infant warmer for postnatal observation and weighing. The infant was placed in the infant warmer until completion of the operation. The parturient women and infant thereafter returned separately to the ward, precluding any opportunity to be together and interact in the operating theater.

### 2.7. Delivery Care for Intervention Group

In the second phase, health-care providers were initially trained regarding the implementation of early SSC. Thereafter, subjects were recruited for the intervention group, and early SSC was implemented. If early SSC was not performed, postpartum data were not collected.

### 2.8. Data Collection

There were 4 data collection time points: 1–3 days, 1 month, 2 months, and 4 months after delivery. The first data collection was conducted using a questionnaire written in Swahili. The data to be collected for all the outcomes are listed under Outcomes (Section 2.2). The second and subsequent data collection was conducted by phone. The data to be collected included exclusive breastfeeding practice status, breastfeeding intention, and infant hospitalization for infectious diseases and diarrhea. To avoid mixing the interventions, a 1-week interval was set after the first data collection for the control group before the start of the second phase (i.e., simulation training for health-care providers).

### 2.9. Instruments

An original 3-choice scale was used for assessing the exclusive breastfeeding practice status, as follows: “Exclusive breastfeeding”, “Breastfeeding plus supplementation”, and “Not breastfeeding”. The 2 options were “Yes” or “No” for breastfeeding intention and infant hospitalization for infectious diseases and diarrhea. The BSS-RI and visual analogue scales, which have been evaluated for reliability and validity, were used for assessing birth satisfaction and perioperative pain. The BSS-RI was incorporated into the 2014 United Kingdom Maternity Survey and the survey was conducted in 4021 women; the mean total BSS-RI score was 8.24 (SD 2.86). Cronbach’s alpha of the BSS-RI total scale was 0.77 [20]. Bahreini et al. assessed the agreements among the visual analogue scale (VAS), color analogue scale (CAS), and verbally administered numeric rating scale (NRS) in the emergency unit setting [21]. They showed that VAS, CAS, and NRS can be reliably and validly used for acute pain measurement in adult patients. A 7-item questionnaire was created to verify the effectiveness of education about the importance of breastfeeding and early SSC for pregnant women using an educational pamphlet. The test has 2 choices, and a higher score meant a higher knowledge level. The education and items on the test were reviewed for content in a previous pilot study [17].

### 2.10. Statistical Analysis

The baseline characteristics are expressed as means (SD) or percentages as appropriate (Table 1). Basic statistical analysis was performed on maternal factors as well as neonatal information by weeks of gestation, newborn’s weight, and Apgar score. Differences between the baseline characteristics in the intervention and control groups were tested with χ^2^ and *t*-test. All statistical inferences were made with a two-sided significance level of 0.05 and performed using SPSS (version Statistics 27; IBM: Armonk, NY, USA, 2020).

The χ^2^ test was used to compare exclusive breastfeeding rates at 4 months after delivery and breastfeeding intention between the 2 groups. Logistic regression analysis was used for adjusted baseline covariates (age, gestational age, parity, education level, occupation, religion, marital status, type of CS). Cox proportional hazard models and log-rank tests were used for the incidence of breastfeeding interruptions and infant hospitalization. The χ^2^ test and linear regression were performed for knowledge tests, perioperative pain visual analogue scale scores, and BSS-RI scores. As a subgroup analysis of all secondary outcomes, comparisons were made for planned CS or emergency CS and primiparas or multiparas.

### 2.11. Ethics Considerations

This study was conducted with strict adherence to the principles of voluntariness, anonymity, harmlessness, and protection of privacy and personal information. Ethics clearance and permission were obtained from the authors’ university ethics review boards and National Institute of Medical Research. The study was pre-registered in the Clinical Trials Registry of the University Hospital Medical Information Network in Japan.

## 3. Results

### 3.1. Demographics and Intervention in Practice

This study involved 172 parturient women who underwent CS: 86 in the intervention group and 86 in the control group. The flow diagram of the intervention and measurement time points for the participants is shown in Figure 1. The baseline characteristics of each group are shown in Table 1, which include age, gestational age, parity, education level, occupation, religion, marital status, type of CS, newborn’s weight, and Apgar score. There were no significant differences in the baseline characteristics between the two groups.

Regarding the practicalities of implementing early SSC, a timing of “within 10 minutes” for receiving SSC was implemented in 81 (94.2%) after primary examination and weighing of the infant (within 10 min), and in 5 (5.8%) after resuscitation. The usual duration of SSC was 20–29 min in 32 (37.2%) participants, followed by 30–39 min in 22 (25.6%) participants, 10–19 min in 17 (19.8%) participants, 40–49 min in 13 (15.1%), and there was one each for 0–10 min and 50–59 min.

### 3.2. Exclusive Breastfeeding Rate at 4 Months after Delivery

There were 57 (76.0%) participants in the intervention group and 58 (76.3%) participants in the control group who answered that they exclusively breastfed their babies at 4 months after delivery. There were no significant differences in the exclusive breastfeeding rates between the two groups (Table 2). In the subgroup analysis under emergency CS, 25 (83.3%) participants in the intervention group and 27 (84.4%) participants in the control group reported exclusive breastfeeding. This was the highest breastfeeding rate, >80%, among the subgroup analyses. Subgroup analysis by type of CS showed no significant difference in the exclusive breastfeeding rates between the two groups (Table 2).

The results of the logistic regression analysis are shown in Table 3. For emergency CS, the adjusted odds ratio for exclusive breastfeeding was 3.206 (95% CI: 1.205–8.532, *p* = 0.020), with planned CS as reference. The adjusted odds ratio in the intervention group was 1.074 (95% CI: 0.477–2.416), with the control group as reference. The association between group differences and exclusive breastfeeding at 4 months was not significant (*p* = 0.864).

### 3.3. Birth Satisfaction Scale—Revised Indicator Score

Higher BSS-RI scores indicate a higher birth satisfaction, with 2 points for each of the 6 questions, for a total score of 12 points. Cronbach’s alpha from this questionnaire was 0.662. The mean score was 8.36 (range 4–12, SD 2.65) for the intervention group and 7.74 (range 3–12, SD 2.04) for the control group (Table 4). Since the histograms were not considered to be normally distributed, the Mann–Whitney U test was used. There was no significant difference in the BSS-RI scores between the two groups (*p* = 0.121). In the subgroup analysis by parity, for multiparas, the mean score was 8.51 (range 4–12, SD 2.75) for the intervention group and 7.66 (range 4–12, SD 1.93) for the control group. The intervention group showed a significantly higher mean score than the control group for multiparas (*p* = 0.03) (Table 4). By type of CS, for emergency CS, the mean score was 7.91 (range 4–12, SD 2.42) for the intervention group and 7.18 (range 3–12, SD 2.02) for the control group. The intervention group showed a significantly higher mean score than the control group for emergency CS (*p* = 0.007) (Table 4). The standardized regression coefficient for BSS-RI scores was 0.194 (95% CI: 0.09–1.545, *p* = 0.028) for emergency CS with planned CS as reference.

For each question, significant differences were found for Q1 (I was not distressed at all during labor) and Q6 (The staff communicated well with me during labor). The mean score of Q1 was 1.40 points (SD 0.73) for the intervention group and 1.29 points (SD 0.64) for the control group (χ^2^ = 6.987, *p* = 0.03). The mean score forof Q6 was 2 points for the intervention group and 1.85 points (SD 0.45) for the control group (χ^2^ = 10.625, *p* = 0.014). In both Q1 and Q6, the intervention group had significantly higher scores than the control group.

### 3.4. Probability of Infant Survival Prior to Hospitalization Owing to Infectious Diseases and Diarrhea

There was one case of infant death at 1–3 days after delivery in the control group. The cause of death was unknown. There were no further cases of infant death.

A Kaplan–Meier survival curve was generated by performing a log-rank test on the infants’ hospitalization owing to infectious diseases and diarrhea. The survival probability for infant hospitalization owing to infectious diseases and diarrhea until 4 months after delivery was 96.4% for the intervention group and 86.4% for the control group (χ^2^ = 3.231, *p* = 0.072). The difference was not significant.

In the subgroup analysis by parity, for multiparas, the probability of survival prior to infant hospitalization owing to infectious diseases and diarrhea or before death until 4 months was 98.5% for the intervention group and 88.3% for the control group (χ^2^ = 5.231, *p* = 0.022) (Appendix B, Figure A1). The difference between the two groups was significant. For all factors, the hazard ratio for infant hospitalization owing to infectious diseases and diarrhea during 4 months of follow-up was not significant.

## 4. Discussion

### 4.1. High Exclusive Breastfeeding Rate at 4 Months after Delivery

Although it was not possible to show that early SSC after CS was associated with significantly higher exclusive breastfeeding rates at 4 months after delivery, thus not fully reflecting the primary outcome, the exclusive breastfeeding rates at 4 months after delivery were high: 76.0% vs. 76.3% in the intervention and control groups. The Tanzania Demographic and Health Survey and Malaria Indicator Survey 2015–2016 reported an exclusive breastfeeding rate of 27% at 4–5 months after birth [5]. The exclusive breastfeeding rates of both groups in the present study were higher than this national average.

A multicenter prospective cohort study conducted in China showed that the exclusive breastfeeding rates at 4 months were 47.39% for mothers who gave birth by vaginal delivery and 32.2% for those who delivered by CS [22]. The breastfeeding rate at 4 months after delivery was notably higher in the present study than in previous studies.

Breastfeeding education during pregnancy may be a factor explaining why approximately 76% of the mothers who gave birth by CS were able to continue exclusive breastfeeding for up to 4 months after delivery, considering that CS is a more difficult condition from which to continue breastfeeding than vaginal delivery. A systematic review of studies on knowledge, attitudes, and practice of exclusive breastfeeding among mothers in East Africa, including Tanzania, revealed that 42.1% of mothers disagreed and 24.0% strongly disagreed that giving breast milk for a newborn immediately and within an hour is important [23]. The review indicated that most mothers also had inadequate knowledge of feeding duration, colostrum, breastfeeding on-demand, benefits of breastfeeding to mothers and babies, and the danger of bottle feeding [23]. A previous systematic review of breastfeeding education showed that health-care providers play a pivotal role in educating and encouraging mothers to start and continue breastfeeding [24]. In the present study, breastfeeding education at the antenatal clinic by a midwife included detailed information on why exclusive breastfeeding is important up to 6 months and what exactly should be fed to the infant. Such breastfeeding education may have contributed to the high exclusive breastfeeding rate.

It was suggested that emergency CS results in a higher rate of exclusive breastfeeding rate than planned CS. This result resembles those of previous studies from other countries. A study examining the effects of CS on breastfeeding in Italy showed that the exclusive breastfeeding rates at 3 months were 66.9% for vaginal delivery, 55.4% for emergency CS, and 55.1% for planed CS [25]. Regarding the impact of CS on breastfeeding in Canada, women who delivered with a planned CS were more likely (OR = 1.61; 95% CI: 1.14, 2.26; *p* = 0.014) to discontinue breastfeeding before 12 weeks postpartum compared with those who had vaginal delivery and emergency CS (adjusted OR = 1.22) [26]. The detailed reasons for choosing exclusive breastfeeding by mothers who had emergency CS was not investigated, and thus we hope to clarify the possible reasons in the future.

Although not significantly different, the slightly higher exclusive breastfeeding rate in the control group than in the intervention group may be due to the season in which an infant was born. A previous study investigated the impact of various factors on exclusive breastfeeding on the basis of Demographic and Health Survey of Tanzania data. Infants born in the short rainy season (April through July) were about twice (AOR 1.79, 95% CI: 1.21–2.65) as likely to be exclusively breastfed as infants born in the dry season (August–November) [27]. The authors described that the short rainy season (April–July) is a period where many parts of Tanzania have abundant food compared with the dry season (August–November). Breastfeeding mothers are more likely to have balanced diets, enabling better production of breast milk [27]. In the present study, data were collected from the middle of June to early July for the control group and from late July to the end of August for the intervention group, and thus the possibility of seasonal effects cannot be ruled out.

### 4.2. Effectiveness of SSC for Birth Satisfaction

The subgroup analysis for multipara or emergency CS found that the intervention group had a significantly higher birth satisfaction score than the control group. A qualitative study investigating mothers’ perceptions of breastfeeding difficulties after CS noted that in most cases, the baby was swaddled in wraps and contact between mother and baby was compromised by theater drapes, baby wraps and the conduction of routine procedures [28]. The study reported the disappointment of mothers in that they had expected to be able to touch their babies as soon as they were born but were not able to, and that they had been informed that SSC was good for breastfeeding but it was not performed [28]. A population-based cohort study of 16,000 women in Sweden showed that emergency CS was significantly related to dissatisfaction with childbirth [29]. However, in the present study, the early SSC after the CS program was thought to increase the level of satisfaction even in an emergency situation and a highly anxious state. A previous study showed that the difference in birth satisfaction between women with and without SSC was particularly large for operative births, particularly for CS [30]. In the present study, the early SSC after CS program similarly contributed to increasing birth satisfaction. Thus, despite the difficulties and distress reported by the mothers who underwent CS, the intervention group had a higher birth satisfaction score than the control group for the multipara and emergency CS group in this study. This suggests that the SSC after the CS program compensated for the disadvantages associated with CS or emergency situations, and contributed to birth satisfaction.

### 4.3. Decrease in the Number of Infants Hospitalized Owing to Infectious Diseases and Diarrhea

In 2020, the infant mortality rate was 35 per 1000 live births in Tanzania [31]. Although the mortality rate in Tanzania has been decreasing year by year, it has remained high compared with other countries. A previous study analyzing trends in neonatal, post-neonatal, and infant mortalities in Tanzania indicated that mothers who gave birth by CS reported higher risks of neonatal, post-neonatal, and infant mortalities [32]. A previous study of risk factors for mortality among Tanzanian infants and children identified respiratory diseases as the most common cause of mortality [33]. Even in the midst of such a situation, the survival probability for infants hospitalized up to 4 months after birth was significantly higher in the intervention group than in the control group for multiparas.

Reports about the intestinal microbiota of very low-birth-weight infants show that dysbiosis is associated with a number of physical and behavioral problems, including autism spectrum disorders, allergy and asthma, gastrointestinal disease, obesity, depression, and anxiety. Additionally, dysbiosis may be prevented or ameliorated in part by prenatal care, breastfeeding, SSC, use of antibiotics only when necessary, and vigilance during infancy and early childhood [34]. Although more research is needed to determine whether similar effects can be seen in normal-birth-weight infants, it is possible that these effects can be achieved through early SSC. In the present study, the lower number of infant hospitalizations from infectious diseases and diarrhea in the intervention group of multiparas was very significant. Both groups had a total fertility rate of approximately two, and it was expected that multiparas would have several children at home. In such a situation, the significant reduction in the hospitalization of recently born infants owing to infectious diseases and diarrhea was a beneficial result.

### 4.4. Future Prospects for Early SSC after CS

This study showed that early SSC after CS was feasible regardless of the type of CS. The results suggest that the procedure can be performed by devising the timing, except in cases where the condition of the mother and infant precludes early SSC. Although previous studies have shown that early SSC was less common in CS than in vaginal delivery, we believe that the application of early SSC after CS will increase if the methods for conducting SSC are devised as in this study.

A meta-analysis of the effects of SSC on the success and duration of the first breastfeeding showed that mother-infant SSC increases the success rate and duration of the first breastfeeding, and hence SSC can be considered an optimal promoter of postnatal care for infants [35]. Many studies have also indicated that early initiation of breastfeeding after delivery had an impact on breastfeeding continuity. One way to increase the exclusive breastfeeding rate as an SSC effect is to encourage breastfeeding during SSC. When training health-care providers, it is necessary to teach them how to safely assist with first-time breastfeeding after CS.

### 4.5. Strengths, Weaknesses, Limitations, and Further Research

The novelty of this study is that it examines the effect of early SSC after CS, which has rarely been implemented in Tanzania. Furthermore, the positive effects of early SSC programs after CS on both mothers and infants is an important outcome. Thus, this program has real potential for promoting infant health in Africa. A previous study suggested that one of the key strategies for reducing maternal and neonatal mortality was to increase institutional delivery and woman-friendly care [36]. According to another study, as respectful newborn care is difficult to define and consequently measure, agreeing on measurable indicators that make sense to women, such as zero separation, SSC, breastfeeding support, and delayed bathing for 24 h, was suggested [15]. An early SSC after CS program may positively contribute to the prevention of “disrespectful maternity care” in facility-based childbirths. We were also able to show that early mother–infant SSC is feasible in a developing country, even during CS. However, further study is needed to determine whether it is feasible to expand this early SSC after CS program to other regions in Tanzania.

This study was a comparison within one tertiary-level facility. Because of the nature of the early SSC after CS program, it was not possible to match the timing of the intervention and data collection for the two groups. As the relationship between the season and the breastfeeding implementation has been reported in a previous study [27], the possibility of seasonal differences in data collection cannot be ruled out. In subsequent studies, it is necessary to devise a method in which data from the two groups are collected simultaneously. As the outcome measures were self-reported by mothers, this may also have influenced the results. The use of more objective measures to evaluate the results would be ideal. Additionally, the facilities investigated in this study were higher tertiary care facilities in Tanzania and were medical institutions with special characteristics. The implementation of future research needs to include different types of facilities in a community.

## 5. Conclusions

Although it was not possible to show that the early SSC after CS program was associated with significantly higher exclusive breastfeeding rates at 4 months after delivery, the rates were higher than those of previous studies and the Tanzanian national average. Specifically, the mean BSS-RI score was significantly higher in the mothers who participated in the early SSC after CS program than in the mothers who did not participate the early SSC after CS program. Moreover, the survival probability for infant hospitalization due to infectious diseases and diarrhea up to 4 months after birth was significantly higher in the intervention group than the control group for multiparas. SSC may also have a positive effect on birth satisfaction and in reducing infant hospitalization from infectious diseases and diarrhea.

## Figures and Tables

**Figure 1 ijerph-20-05772-f001:**
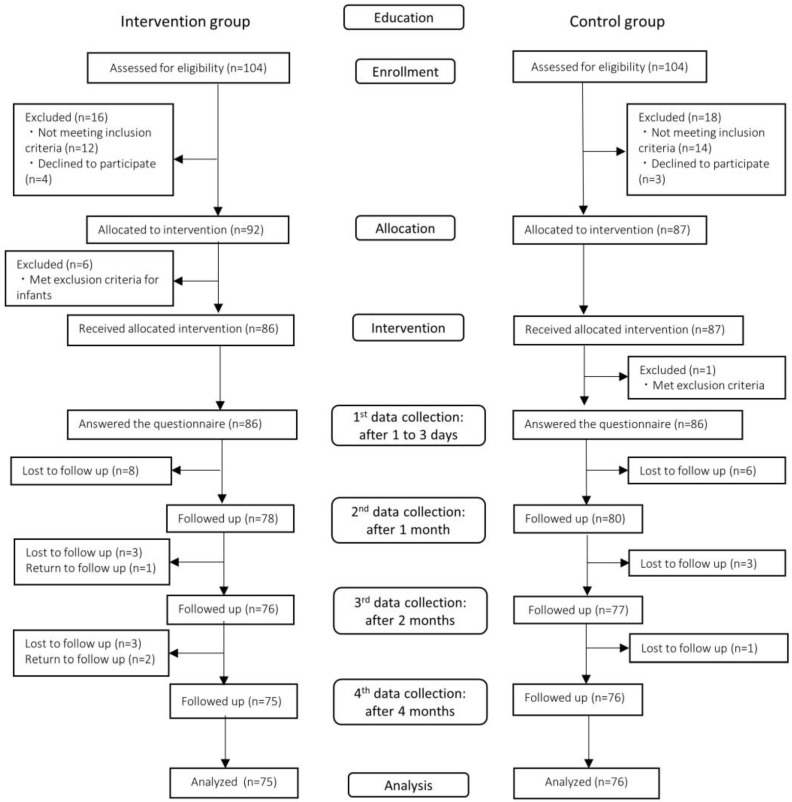
Flow diagram of participants.

**Table 1 ijerph-20-05772-t001:** Characteristics of participants.

	Intervention (n = 86)		Control (n = 86)			
	n	%	n	%	χ^2^	*p*-Value
**Age (years)**						
18–19	1	1.2	0	0	2.96	0.397
20–29	21	24.4	29	33.7		
30–39	60	69.8	52	60.5		
40–43	4	4.7	5	5.8		

**Parity**						
Primipara	21	24.4	16	18.6	0.861	0.353
Multipara	65	75.6	70	81.4		

**Gestation age**						
Preterm	9	10.5	18	20.9	3.559	0.059
Term	77	89.5	68	79.1		

**Education**						
Primary	10	11.6	16	18.6	1.71	0.425
Secondary	27	31.4	23	26.7		
Diploma <	47	54.7	46	53.5		
Unclear	2	2.3	1	1.2		

**Occupation**						
House wife	11	12.8	15	17.5	1.42	0.492
Entrepreneur	33	38.4	34	39.5		
Employee	42	48.8	34	39.5		
Unclear	-	-	3	3.5		

**Religion**						
Christian	57	66.3	61	70.9	0.432	0.511
Muslim	29	33.7	25	29.1		

**Marital status**						
Married	77	89.5	79	91.9	0.276	0.6
Un married	9	10.5	7	8.1		

**Type of CS**						
Planned	51	59.3	50	58.1	0.24	0.877
Emergency	35	40.7	36	41.9		

**Newborn’s weight ***						
1999 g >	1	1.1	1	1.1	0.168	0.983
2000–2999 g	29	33	31	35.2		
3000–3999 g	50	56.8	48	54.5		
4000 g ≦	8	9.1	7	8		
Unclear	-	-	1	1.1		

**Apger score**						
After 1 min						
7 ≧	10	11.4	12	13.6	0.329	0.566
8 ≦	78	88.6	72	81.8		
Unclear	-	-	4	4.5		

***** Two sets of twins were included. The total is 196 infants, 88 in the control group and 88 in the intervention group.

**Table 2 ijerph-20-05772-t002:** Comparisons of breastfeeding status by parity and cesarean section at 4 months after delivery.

		Intervention (n = 75)	Control (n = 76)	χ^2^	*p*-Value
		n (%)	n (%)		
	Exclusive breastfeeding	57 (76.0)	58 (76.3)	0.002 *	1
	Others	18 (24)	18 (23.7)		

**Type of CS**	**Planned (n = 99)**	Intervention (n = 45)	Control (n = 44)	χ^2^	*p*-value
		n (%)	n (%)		
	Exclusive breastfeeding	32 (71.1)	31 (70.5)	0.005 *	1
	Others	13 (28.9)	13 (29.5)		

	**Emergency (n = 62)**	Intervention (n = 30)	Control (n = 32)	χ^2^	*p*-value
		n (%)	n (%)		
	Exclusive breastfeeding	25 (83.3)	27 (84.4)	0.012*	1
	Others	5 (16.7)	5 (15.6)		

* Expectation is less than 5, using Fisher’s direct method. CS: Caesarean section.

**Table 3 ijerph-20-05772-t003:** Logistic regression analysis results for exclusive breastfeeding rate at 4 months.

	Adjusted	95%CI		*p*-Value
	OR	Lower	Upper	
**Age**	1.045	0.953	1.147	0.347
**Parity**				
Primipara	1			
Multipara	1.866	0.604	5.768	0.278

**Gestation age**				
Preterm	1			
Term	1.251	0.337	4.636	0.738

**Education**				
Primary	1.144	0.278	4.713	0.852
Secondary	1.632	0.553	4.814	0.375
Diploma<	1			

**Occupation**				
House wife	2.590	0.558	12.028	0.225
Entrepreneur	1.674	0.661	4.238	0.277
Employee	1			

**Religion**				
Christian	1			
Muslim	1.224	0.480	3.120	0.672

**Marital status**				
Married	1.752	0.437	7.030	0.429
Un married	1			

**Type of CS**				
Planned	1			
Emergency	3.206	1.205	8.532	0.020

**Group**				
Intervention	1.074	0.477	2.416	0.864
Control	1			

**Table 4 ijerph-20-05772-t004:** Comparisons of birth satisfaction outcomes by parity and type of caesarean section.

		Intervention (n = 86)			Control (n = 86)			
		Mean	Range	SD	Mean	Range	SD	*p*-value
	The Birth Satisfaction (0–12)	8.36	4–12	2.65	7.74	3–12	2.04	0.121

**Parity**	**Primipara**	Intervention (n = 21)			Control (n = 16)			
		Mean	Range	SD	Mean	Range	SD	*p*-value
	The Birth Satisfaction (0–12)	7.9	4–12	2.32	8.06	3–12	2.46	0.575

	**Multipara**	Intervention (n = 65)			Control (n = 70)			
		Mean	Range	SD	Mean	Range	SD	*p*-value
	The Birth Satisfaction (0–12)	8.51	4–12	2.75	7.66	4–12	1.94	0.030

**Type of CS**	**Planned CS**	Intervention (n = 51)			Control (n = 50)			
		Mean	Range	SD	Mean	Range	SD	*p*-value
	The Birth Satisfaction (0–12)	8.65	4–12	2.78	8.15	4–12	1.98	0.724

	**Emergency CS**	Intervention (n = 35)			Control (n = 36)			
		Mean	Range	SD	Mean	Range	SD	*p*-value
	The Birth Satisfaction (0–12)	7.91	4–12	2.42	7.18	3–12	2.02	0.007


CS = Caesarean section.

## Data Availability

The data presented in this study are available on request from the corresponding author. The data are not publicly available due to privacy.

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
