# Peer review of "Effectiveness of an Early Skin-to-Skin Contact Program for Pregnant Women with Cesarean Section: A Quasi-Experimental Trial"

_ijerph, 2023, doi:10.3390/ijerph20105772_

Round 1
Reviewer 1 Report
This study in single centre in Tanzania aimed to clarify the effectiveness of an early skin-to-skin contact 11 (SSC) after cesarean section (CS) program. This study involved 172 parturient women who underwent CS, with 86 in the intervention group and 86 in the control group, so numbers of participants are small.
This study found that early SSC after CS program showed a positive effect on the birth satisfaction of women undergoing emergency but not planned CS. It also reduced the incidence of infants hospitalised owing to infectious diseases and diarrhea for multiparas.
My questions:
1."The primary outcome was the rate of exclusive breastfeeding at 4 months after delivery". Why you chose only 4 months not 6 or even 9-12 month after delivery? Please clarify this methodology.
2. line 86 word babies missing spelling.
3. Methods are explained very clearly.
4. Results are properly showed and conclusions are correct.
But I'm not sure are these conclusions have some kind of novelty in the subject.
Author Response
Reviewer 1
Comments
1."The primary outcome was the rate of exclusive breastfeeding at 4 months after delivery". Why you chose only 4 months not 6 or even 9-12 month after delivery? Please clarify this methodology.
Response:
Thank you for pointing out this point. According to Tanzania Demographic and Health Survey, the exclusive breastfeeding rate dropped sharply from 4 months postpartum. Therefore, we thought it was important to first aim to improve the exclusive breastfeeding rate at 4 months. We have added this in the text. We would also like to conduct research on the effectiveness of continuing the program for 6 months and 1 year in the future.
(Page 8, lines 13-16)
2.2 Outcomes
The primary outcome was the rate of exclusive breastfeeding at 4 months after delivery. According to Tanzania Demographic and Health Survey, the percentage of exclusive breastfeeding dropped sharply from 4 months postpartum [5]. It was important to first aim to improve the breastfeeding rate at 4 months.
- line 86 word babies missing spelling.
Response:
We apologize very much for the type miss. It has been corrected.
(Page 7, lines 8-10)
A previous study has reported that CS-delivered babies had a 98% decrease in the odds (aOR = 0.02, 95% CI: 0.01-0.05) of receiving SSC compared with vaginal-delivered babies [15].
- Methods are explained very clearly.
Response:
We appreciate your taking the time to review our manuscript and provide us with warm comments.
- Results are properly showed and conclusions are correct.
But I'm not sure are these conclusions have some kind of novelty in the subject.
Response:
Thank you for your very important input. The novelty of this study is that, although many studies have been conducted and reported on early SSC during vaginal delivery in developing countries, we believe that it was significant to examine the effect of early SSC during CS. It is also very significant that we were able to examine not only breastfeeding as an outcome, but also the effects on infant infection and diarrhea, and to report positive results. The novelty of this study has been added in the text.
(Page 29, lines 5-7)
4.5 Strengths, weaknesses, limitations, and further research
The novelty of this study is that it examines the effect of early SSC after CS, which has rarely been implemented in Tanzania. Furthermore, the positive effects of early SSC programs after CS on both mothers and infants is an important outcome. Thus, this program has real potential for promoting infant health in Africa.

Reviewer 2 Report
The authors conducted a quasi-experimental study to investigate the effectiveness of an early skin-to-skin contact program for pregnant women with cesarean section. I congratulate the authors for choosing an essential topic. This study should raise awareness of the skin-to-skin contact benefits. The scenario training for healthcare providers addition was an excellent idea. Moreover, I would like to highlight the investigation of this topic in a third-world country. I truly have a few spell-checks to raise. Otherwise, I would like to wish all the luck to the authors. You should investigate this topic further and try to implement the program you described in detail. The program has the potential to promote mothers' and infants' health in Africa.
The introduction provides sufficient data regarding the consequences of emerging cesarean section rates, and the importance of early breastfeeding. The authors provided materials and methods to an extensive degree. The results are presented clearly, and the conclusions are consistent with the evidence and arguments presented.
Figure 1 is missing, and You should upload it in the final version of the manuscript.
Author Response
Reviewer 2
Comment
The authors conducted a quasi-experimental study to investigate the effectiveness of an early skin-to-skin contact program for pregnant women with cesarean section. I congratulate the authors for choosing an essential topic. This study should raise awareness of the skin-to-skin contact benefits. The scenario training for healthcare providers addition was an excellent idea. Moreover, I would like to highlight the investigation of this topic in a third-world country. I truly have a few spell-checks to raise. Otherwise, I would like to wish all the luck to the authors. You should investigate this topic further and try to implement the program you described in detail. The program has the potential to promote mothers' and infants' health in Africa.
The introduction provides sufficient data regarding the consequences of emerging cesarean section rates, and the importance of early breastfeeding. The authors provided materials and methods to an extensive degree. The results are presented clearly, and the conclusions are consistent with the evidence and arguments presented.
Figure 1 is missing, and You should upload it in the final version of the manuscript.
Response:
We appreciate your taking the time to review our manuscript and provide us with warm comments. I am sorry for the shortage of the figure. After reconsideration, we have removed Figure 1 as we believe the figure is unnecessary. Figure 2 has been changed to Figure 1.
(Page12, line14)
2.6 Delivery care for control group
This study was divided into 2 phases (Figure 1).
(Page17, lines1-5)
3 Results
3.1 Demographics and intervention in practice
This study involved 172 parturient women who underwent CS; 86 in the intervention group and 86 in the control group. The flow diagram of the intervention and measurement time points for the participants is shown in Figure 1.

Reviewer 3 Report
A very important issue in terms of infant health has been studied. The material and work plan are well presented in the article. However, there are deficiencies in statistical analyzes and findings.
Have the data been tested for normal distribution (eg. birth satisfaction scores)? how the analyzes were performed in logistic analysis for the unclear data (as in the education and occupation variable).
There is 1 person under the age of 20; age limits can be arranged as under 30 and over
It is not clear how birth weights are separated. Compatibility of the weight according to the week of gestation should be checked (>10 p)
Author Response
Reviewer 3
Comment
A very important issue in terms of infant health has been studied. The material and work plan are well presented in the article. However, there are deficiencies in statistical analyzes and findings.
Have the data been tested for normal distribution (eg. birth satisfaction scores)? how the analyzes were performed in logistic analysis for the unclear data (as in the education and occupation variable).
There is 1 person under the age of 20; age limits can be arranged as under 30 and over
It is not clear how birth weights are separated. Compatibility of the weight according to the week of gestation should be checked (>10 p)
Response:
Thank you for your very important comments. We appreciate your taking the time to review our manuscript and provide us with helpful feedback. We have carefully reviewed and corrected the areas you pointed out.
- Have the data been tested for normal distribution (eg. birth satisfaction scores)? how the analyzes were performed in logistic analysis for the unclear data (as in the education and occupation variable).
Response:
Thank you for your important question. The test was done after checking if the distribution is normal or not. I have added it to the text.
In the logistic regression analysis, comparisons were made in the categories shown in Table 3. For example, for education, the odds ratios for primary and secondary education are based on the final education being a diploma.
(Page 21, Lines 5-6)
3.3 Birth Satisfaction Scale-Revised Indicator score
Higher BSS-RI scores indicate a higher birth satisfaction, with 2 points for each of the 6 questions, for a total score of 12 points. The Cronbach’s alpha from this questionnaire was 0.662. The mean score was 8.36 (range 4-12, SD 2.65) for the intervention group and 7.74 (range 3-12, SD 2.04) for the control group (Table 4). Since the histograms were not considered to be normally distributed, the Mann-Whitney U test was used. There was no significant difference in the BSS-RI scores between the 2 groups (p = .121).
- There is 1 person under the age of 20; age limits can be arranged as under 30 and over
Response:
Thanks for pointing this out. We agree that age division is a very important factor. Although one participant in the intervention group was under 20 years of age, in this study, Tanzanians 18 years of age and older are considered adults, and it is unlikely that there is a significant difference in child-rearing ability. Therefore, we did not consider this to be a difference that would have a significant impact on our study. In addition, there was no significant difference between the two age groups of 30 years old and above, so we have used the current age division.
- It is not clear how birth weights are separated. Compatibility of the weight according to the week of gestation should be checked (>10 p)
Response:
Thank you for your important remarks. As you said, it is important to confirm whether the infant is a light for gestational age infant, a large for gestational age infant, or an appropriate for gestational age infant. We did not collect that data this time, so this will be an area for improvement next research. We analyzed basic statistics by weight and by weeks of gestation. Table 1 shows newborn’s weight separated by every 1000g for convenience to facilitate comparison of the two groups. Since there was no difference between the two groups when the test was conducted separately for low birthweight infants, normal infants, and large infants, we have used this expression.
(Page 14, Lines 14-15)
2.10 Statistical analysis
The baseline characteristics were expressed as mean (SD) or percentage as appropriate (Table 1). Basic statistical analysis was performed on maternal factors as well as neonatal information by weeks of gestation, newborn’s weight, and Apgar score. Differences between the baseline characteristics in the intervention and control groups were tested with χ2 and t-test. All statistical inferences were made with a two-sided significance level of .05 and performed using SPSS version 27.
